# The Cytokine Nicotinamide Phosphoribosyltransferase (eNAMPT; PBEF; Visfatin) Acts as a Natural Antagonist of C-C Chemokine Receptor Type 5 (CCR5)

**DOI:** 10.3390/cells9020496

**Published:** 2020-02-21

**Authors:** Simone Torretta, Giorgia Colombo, Cristina Travelli, Sara Boumya, Dmitry Lim, Armando A. Genazzani, Ambra A. Grolla

**Affiliations:** 1Department of Pharmaceutical Sciences, Università del Piemonte Orientale, 28100 Novara, Italy; simone.torret89@gmail.com (S.T.); giorgia.colombo@uniupo.it (G.C.); sara_posta1993@yahoo.it (S.B.); dmitry.lim@uniupo.it (D.L.); armando.genazzani@uniupo.it (A.A.G.); 2Department of Pharmaceutical Sciences, Università di Pavia, 27100 Pavia, Italy; cristina.travelli@unipv.it

**Keywords:** eNAMPT, visfatin, CCR5, antagonism, cancer, calcium signaling, migration

## Abstract

(1) Background: Extracellular nicotinamide phosphoribosyltrasferase (eNAMPT) is released by various cell types with pro-tumoral and pro-inflammatory properties. In cancer, eNAMPT regulates tumor growth through the activation of intracellular pathways, suggesting that it acts through a putative receptor, although its nature is still elusive. It has been shown, using surface plasma resonance, that eNAMPT binds to the C-C chemokine receptor type 5 (CCR5), although the physiological meaning of this finding is unknown. The aim of the present work was to characterize the pharmacodynamics of eNAMPT on CCR5. (2) Methods: HeLa CCR5-overexpressing stable cell line and B16 melanoma cells were used. We focused on some phenotypic effects of CCR5 activation, such as calcium release and migration, to evaluate eNAMPT actions on this receptor. (3) Results: eNAMPT did not induce ERK activation or cytosolic Ca^2+^-rises alone. Furthermore, eNAMPT prevents CCR5 internalization mediated by Rantes. eNAMPT pretreatment inhibits CCR5-mediated PKC activation and Rantes-dependent calcium signaling. The effect of eNAMPT on CCR5 was specific, as the responses to ATP and carbachol were unaffected. This was strengthened by the observation that eNAMPT inhibited Rantes-induced Ca^2+^-rises and Rantes-induced migration in a melanoma cell line. (4) Conclusions: Our work shows that eNAMPT binds to CCR5 and acts as a natural antagonist of this receptor.

## 1. Introduction

Extracellular nicotinamide phosphoribosyltrasferase (eNAMPT), also known as PBEF or visfatin, is the secreted form of NAMPT (EC 2.4.2.12), a key enzyme involved in maintaining the balance of NAD and ATP levels in cells [1]. NAMPT, starting from nicotinamide (Nam), adenosine triphosphate (ATP) and phosphoribosyl pyrophosphate (PRPP), catalyzes the production of nicotinamide mononucleotide (NMN), a key precursor of NAD in mammalian cells. Importantly, different cell types secret NAMPT in the extracellular space, where it is now considered a metabokine with pro-inflammatory and pro-tumoral activity [2,3]. Of great importance, eNAMPT is overexpressed in several disorders, including cancer [4,5] where eNAMPT controls angiogenesis, tumor growth and metastasis [6,7]. The mechanism by which eNAMPT exerts its function is still an open debate. Three theories, not necessarily mutually exclusive, are present in the literature: (i) eNAMPT binds to a yet unidentified receptor; and/or (ii) eNAMPT is enzymatically active in the extracellular milieu and this is fundamental for its actions; (iii) it is carried in the systemic circulation in extracellular vesicles (EV) and is liberated upon internalization, enhancing NAD^+^ biosynthesis [8]. This latter possibility was demonstrated by Yoshida et al. both in mice and humans [8].

The possibility that eNAMPT binds to an extracellular receptor is supported by the observation that stimulation of cancer cells with exogenous eNAMPT in solutions not containing the enzymatic substrates is sufficient to activate specific intracellular signaling pathways (e.g., STAT3, NF-κB, Akt, P38) within minutes, which indicates that it is likely that eNAMPT binds to and activates a cell surface receptor. In this respect, the insulin receptor was initially proposed but the report was retracted [9]. More recently, Camp et al. demonstrated that eNAMPT induces lung inflammation via a direct interaction with Toll-like receptor 4 (TLR4). Moreover, computational analysis demonstrated that eNAMPT and MD-2, a TLR4-binding protein, share ≈30% sequence identity [10]. This interaction was recently confirmed by Managò and colleagues [11].

Less recently, it was reported that eNAMPT is able to selectively inhibit infection of monocytes by human immunodeficiency virus (HIV) and this activity was linked to a direct interaction with the C-C chemokine receptor type 5 (CCR5). The interaction was solely demonstrated by surface plasmon resonance (SPR) with a K_D_ of 5 µM in the 1:1 binding model [12] and no functional studies were performed. 

CCR5 is a seven transmembrane, G-protein coupled receptor (GPCR), involved in inflammatory responses and it also serves as a coreceptor for the entry of R5 strains of HIV [13,14]. It recognizes Rantes (also known as CCL5) as its main ligand, although other proteins are known to affect its function including the agonists CCL3 (also known as MIP-1α), MIP1-β, CCL2, CCL8, CCL11, and CCL14 and the natural antagonist CCL7 (also known as MCP-3) [15]. Maraviroc, a competitive molecular antagonist, is currently used in HIV therapy [16]. CCR5 is overexpressed in several cancers (e.g., breast cancer, melanoma) and it has been suggested that the activation of this receptor controls tumor development [17,18]. For example, in melanoma, CCR5 expression on stromal cells is necessary for the spread of B16 melanoma cells to the lungs [19] and in CCR5-deficient mice B16 melanoma growth is delayed [20].

The present contribution aims at elucidating the correlation between CCR5 and eNAMPT in cancer cells. We now show that eNAMPT binds to CCR5 in cancer cells and acts as a natural antagonist of this receptor. Antagonism of CCR5 alone is unlikely to explain all the actions mediated by eNAMPT [3], and it therefore likely that other eNAMPT receptors exist, including the recently demonstrated TLR4.

## 2. Materials and Methods

### 2.1. Cell Culture

HeLa (human cervix carcinoma) and B16 (murine melanoma) cells were cultured in Dulbecco’s modified Eagle’s medium (DMEM, Sigma-Aldrich, St. Louis, MO, USA)) supplemented with 10% fetal bovine serum (FBS), 2 mg/mL glutamine, 10 U/mL penicillin, and 100 µg/mL streptomycin (Sigma-Aldrich). Cells were maintained in a humidified incubator supplied with 5% CO_2_/95% air at 37 °C and detached by Trysin-EDTA.

### 2.2. Recombinant eNAMPT Preparation 

Wild-type murine full-length and H247E NAMPT (ORF GenBank BC018358) were cloned in pET28a (NdeI/EcoRI). Recombinant eNAMPTs were expressed in *E. coli*. (ClearColi, BL21-DE3), inducted with IPTG 0.5 mM for 3 h at 20 °C, and purified by His-tag affinity chromatography with NiNTA Superflow resin (Qiagen, Hilden, Germany). Protein purity was determined by SDS-page. Only protein preparations with a purity higher than 95% and with low endotoxin levels (<0.01 U, LAL test) were used. eNAMPT in most experiments was used at a concentration of 500 ng/mL, as in most other work on the subject (e.g., [12,21]) except for experiments in which equilibrium or semi-equilibrium conditions were reached.

### 2.3. Generation of Stable HeLa-CCR5 Cell Line

Murine CCR5 was cloned in the pLV lentiviral vector. Correct insertion and sequence were confirmed by DNA sequencing. The lentiviral particles were produced as described elsewhere (Grolla et al., 2015) in HEK293T cells transfected with pMDLg/pRRE, pMD2.VSVG, pRSV-Rev, and pLV-CCR5/pLV-empty plasmids. Briefly, after 48 h, cell medium was collected, filtrated, and centrifuged for 90 min at 100,000 *g*. The viral particles were resuspended and used to infect HeLa cells, after virus titration. Stable scramble (HeLa-SCR) and HeLa-CCR5 were stained with anti-CCR5 PE (Biolegend, San Diego, CA, USA) for 20 min at 4°C, then washed twice in PBS and the relative expression of CCR5 analyzed using by flow cytometry (BD Accuri C6).

### 2.4. Binding Assay 

First, Rantes-PE was prepared by incubating avidin-PE (Invitrogen, Carlsbad, CA, USA) and biotin-Rantes (Chemotactics) for 3 min at RT. Following this, 1 × 10^5^ cells suspension was incubated with 2U/1 × 10^5^ cells of Heparinise I and III (2U/10^6^ cells, Sigma-Aldrich) for 1 h at 4 °C. Then, cells were washed and incubated with eNAMPT (125 µg/mL; 2.25 µM) or maraviroc (10 µM, Sigma Aldrich) at 4 °C in complete medium (200 µL). After 20 min, Rantes-PE (25 ng; 16 nM) was added to the cell suspension for 2 h at 4 °C. Then, cells were washed three times in ice cold PBS, resuspended in FACS Buffer (Hanks’ Balanced Salt solution HBSS + 0.5% BSA), and samples were analyzed by flow cytometry (BD Accuri C6). eNAMPT was used at this high concentration considering both the molecular weight and the oligomerization state of the proteins, to take into account the reported difference in affinity to CCR5 (0.4 nM for RANTES vs. 5 µM for eNAMPT [12,22]. In preliminary experiments, lower concentrations of eNAMPT were devoid of effect.

### 2.5. CCR5 Internalization

First, 2 × 10^5^ cells were plated in 96-well plates and treated with vehicle, Rantes (100 ng/mL; 12 nM Peprotech), CCL3 (100 ng/mL; Peprotech), CCL7 (2.5 µg/mL; Peprotech), maraviroc (10 µM; Sigma-Aldrich), and/or eNAMPT (2.5 µg/mL; 45 nM) for 60 min at 37 °C or 4 °C (as a negative control). Cells were washed twice in PBS and resuspended in 100 μL of phosphate buffer saline (PBS) and stained with anti-CCR5 PE for 20 min at 4 °C. Cells were then washed twice in PBS and cell surface expression of CCR5 was determined with a BD Accuri FACS. In parallel, cells were plated on a coverslip and treated with vehicle, Rantes (100 ng/mL; 12 nM), CCL3 (100 ng/mL; 9.9 nM), and/or eNAMPT (2.5 µg/mL; 45 nM), CCL7 (250 ng/mL; 22 nM) for 60 min at 37 °C or 4 °C. Then, cells were fixed in 4% formaldehyde for 15 min at 4 °C. Subsequently, cells were stained with a primary antibody anti-mouse CCR5-PE and DRAQ5 (Invitrogen). For these experiments we used 12 nM Rantes as this concentration gave a significant level of internalization. We therefore used eNAMPT roughly at a three-fold higher concentration (45 nM). Lower concentrations of eNAMPT did not show a significant effect on preventing Rantes-mediated internalization (e.g., 500 ng/mL = 9 nM eNAMPT did not affect Rantes-mediated internalization).

### 2.6. Fura-2 Imaging

Cells were loaded with 2 μM Fura-2-AM (Invitrogen) in KRB solution (Krebs-Ringer modified buffer: 125 mM NaCl, 5 mM KCl, 1 mM Na_3_PO_4_, 1 mM MgSO_4_, 5.5 mM glucose, 20 mM HEPES, pH 7.4) supplemented with 2 mM CaCl2, 0.01% pluronic acid and 5μM sulfinpyrazone. After washing and de-esterification (30 min), the coverslip was mounted in a chamber and placed on the stage of a Leica epifluorescent microscope equipped with a S Fluor 40×/1.3 objective. Cells were stimulated with the indicated treatments and excited at 340/380 nm by the monochromator Polichrome V (Till Photonics, Munich, Germany) and the fluorescent signal was collected by a CCD camera (Hamamatsu, Japan) through bandpass 510 nm filter; the experiments were controlled and images analyzed with MetaFluor (Molecular Devices, Sunnyvale, CA, USA) software. To quantify the difference in the amplitude of Ca2+ transients, the ratio values were normalized according to the formula (ΔF)/F0 (referred to as normalized (norm.) ratio). NMN, ATP and carbachol were purchased from Sigma-Aldrich.

### 2.7. Western Blot

First, 5 × 10^5^ cells were treated at the indicated time-points and the cells were lysed in 80 μL in lysis buffer composed of 20 mM HEPES, 100 mM NaCl, 5 mM EDTA, 1% Nonidet P-40, and Protease and Phosphatase Inhibitor Cocktail (Sigma). Proteins quantification was performed with BCA Protein Assay (Thermo Fisher Scientific, Waltham, MA, USA), and proteins were resolved on SDS–PAGE. Antibodies used: rabbit polyclonal CCR5 antibody (AB65850 Abcam); rabbit polyclonal antibody anti-p42/44 (Cell Signalling Technology); mouse anti-actin (Sigma); rabbit pPKC (pan) ZT410 Cell signaling and peroxidase-conjugated secondary antibodies (Bio-Rad, Hercules, CA, USA). Densitometry analysis was evaluated using was performed with Quantity One program (Bio-Rad).

### 2.8. Wound-Healing Assay

When confluent monolayers of B16 cells were established, we performed a cross-shaped scratch with tip. Then, the cells were washed twice with PBS to remove residual cell debris. Cells were then incubated with Rantes (100 ng/mL; 12 nM), eNAMPT (500 ng/mL; 9 nM), CCL7 (250 ng/mL; 22 nM), CCL3 (100 ng/mL; 9.9 nM), and maraviroc (10 µM) or the combination for 24 h and pictures of a defined wound spot were made at different time points. The area of the wound in the microscopic pictures was measured and analyzed using Image J software (National Institutes of Health, MD, USA). 

### 2.9. Statistical Analysis 

Statistics were calculated with Graphpad Prism version 6 software. A two-sided unpaired Student’s *t*-test was used to compared unmatched groups and expressed as: t test * *p* < 0.05; ** *p* < 0.01; *** *p* < 0.001. 

## 3. Results

### 3.1. eNAMPT Binds to CCR5, but Does Not Act as an Agonist 

A previous report demonstrated that eNAMPT interacted with CCR5 with a K_D_ of 5 µM in a 1:1 binding model by surface plasmon resonance [21]. 

To understand the biological significance of this interaction, we generated HeLa cells overexpressing murine CCR5 (HeLa-CCR5) and the respective control represented by HeLa cells infected with the empty vector (HeLa-SCR). The relative overexpression of CCR5 was determined by flow cytometry analysis (Figure 1A) and Western blot analysis (Figure 1B). 

As shown in Figure 1B, in HeLa-CCR5 cells the pretreatment with eNAMPT (2.25 µM) reduced the percentage of Rantes-PE (16 nM) positive cells, similarly to maraviroc (10 µM), confirming that eNAMPT binds to CCR5 [21]. Lower concentrations of eNAMPT were unable to elicit an effect, but this is likely to be due to the difference in K_D_ of the two ligands (K_D_ for Rantes approx. 0.4 nM vs. K_D_ for eNAMPT of approx. 5 5 µM; [12,22])

In this report, it was also shown that eNAMPT is capable of inhibiting infections by R5 HIV in monocytes [12]. Given that both CCR5 agonists and antagonists inhibit HIV infections [8], we next analyzed the consequence of such binding. For these experiments, lower doses were used as there was no competition between RANTES and eNAMPT. 

We first evaluated whether eNAMPT parallels the effects of Rantes. As it can be seen in Figure 2A, Rantes (25 ng/mL; 3 nM) was able to elicit a marked ERK phosphorylation in a time-dependent manner in HeLa-CCR5 cells, but not in HeLa-SCR cells. A trend of pERK activation was observable when cells were treated with eNAMPT (250–1000 ng/mL = 4.5 nM–8 nM), although no differences were present between HeLa SCR and CCR5 cells, meaning that it is independent of CCR5 (Figure 2A and Appendix A). The ability of eNAMPT to induce pERK activation, independently on CCR5 pathway, has been reported previously [6,11].

Calcium signaling has also been associated with CCR5 and we therefore also evaluated this pathway. As it can be seen in Figure 2B, Rantes (3 nM) induced a cytosolic Ca^2+^ increase in HeLa-CCR5 (but not in HeLa-SCR; not shown), while eNAMPT (250–1000 ng/mL = 4.5 nM–18 nM) was again unable to do so (Figure 2B, concentrations of 250 and 1000 ng/mL not shown). In support of this data, we investigated also the ability of eNAMPT to induce CCR5 internalization, a common feature of CCR5 agonists. As shown in Figure 2C, both the agonists Rantes (100 ng/mL = 12 nM) and CCL3 (100 ng/mL = 9.9 nM) reduced the surface expression of CCR5 after 60 min. On the other hand, CCL7 (250 ng/mL = 22 nM), maraviroc (10 µM), and eNAMPT (up to 2.5 µg/mL = 45 nM) did not induce internalization of CCR5 (Figure 2C). 

Considering these results, it would appear that eNAMPT is not a CCR5 agonist at nanomolar concentrations.

### 3.2. eNAMPT Antagonizes CCR5 Activation in Hela-CCR5 Cells

We next tested whether instead eNAMPT was able to modulate the responses of agonists on CCR5. 

At first, we evaluated the effect of eNAMPT cotreatment with Rantes in CCR5 internalization. We observed that eNAMPT, similarly to CCL7 and maraviroc, reduced Rantes-mediated CCR5 internalization (Figure 3A,B). We corroborated this evidence by immunofluorescence, in which eNAMPT alone did not induce CCR5 internalization, while it prevented Rantes-mediated internalization (Appendix A).

We then investigated the downstream pathways of CCR5. We found that neither eNAMPT co-treatment nor pretreatment was able to interfere with Rantes-mediated ERK activation, while, as expected, this pathway was antagonized by maraviroc (Appendix A). On the contrary, eNAMPT was able to antagonize CCR5-mediated phosphorylation of PKC induced by Rantes and CCL3 in Hela-CCR5 cells. As shown in Figure 3C, the known antagonist CCL7, eNAMPT and, as expected, CCL3 and Rantes all induced PKC phosphorylation. On the other hand, pretreatment of eNAMPT prevented both PKC activation induced by Rantes and CCL3, similarly to CCL7 (Figure 3C). All this evidence suggested that eNAMPT might act as an antagonist on some of the effects mediated by CCR5, probably in a time-dependent manner. 

Observing the antagonistic effect of eNAMPT on PKC activation mediated by DAG, we next tested whether eNAMPT was able to modulate CCR5-mediated calcium signaling. In accordance with our hypothesis, preincubation with eNAMPT (500 ng/mL = 9 nM) reduced significantly Rantes-induced cytosolic Ca^2+^-rises as determined by the area under the curve (AUC) of the Ca^2+^-rise or the percentage of responding cells (Figure 4A). The effect of eNAMPT was quantitatively similar to that of maraviroc (Figure 4A). It is important to stress that the effect of eNAMPT could not be mimicked by the buffer in which the protein was dissolved, or by proteins isolated in the same manner, such as NMNAT and ADH, at identical concentrations (Appendix A). To evaluate whether the effect of eNAMPT on calcium signaling was specific for CCR5, we evaluated its effect on Ca^2+^ signaling induced by ATP or carbachol (CCh), whose receptors are present both in HeLa-SCR and in HeLa-CCR5 cells. As it can be observed in Figure 4B,C, eNAMPT (500 ng/mL = 9 nM) did not affect the calcium responses to ATP or CCh in either HeLa-SCR or HeLa-CCR5 cells. All this data suggests that eNAMPT selectively antagonizes Rantes-dependent calcium signaling. Very recently, Sayers et al., demonstrated a modulation of calcium release mediated by eNAMPT, which, as a dimer, increased glucose-stimulated intracellular calcium levels [23]. In this respect, in our model we were unable to appreciate any effect of eNAMPT alone at the concentrations tested (500–1000 ng/mL = 9–18 nM).

### 3.3. eNAMPT Modulation of Rantes-Mediated Calcium Signaling is Independent on Its Enzymatic Activity

To test if the reduction of Rantes-dependent calcium mobilization by eNAMPT was mediated by its enzymatic activity, we investigated the effect of a mutated and enzymatically inactive form of eNAMPT (H247E; [24]). The preincubation with eNAMPT H247E (500 ng/mL; 9 nM) reduced the calcium mobilization induced by Rantes in the same manner to the WT enzyme (Figure 5A). In support of this, preincubation with nicotinamide mononucleotide (NMN), the product of eNAMPT, was devoid of any effect on calcium mobilization induced by Rantes (Figure 5B). 

These data suggest that the reduction in calcium signaling by eNAMPT is mediated by the physical interaction with CCR5 and that the enzymatic activity is dispensable.

### 3.4. eNAMPT Modulates Rantes-Mediated Calcium Signaling and Migration in Melanoma Cells 

We next investigated whether the interaction between CCR5 signaling and eNAMPT could be evaluated also in a system in which CCR5 was present endogenously. In this respect, we chose a murine melanoma cell line, B16, in which we have previously shown an effect of eNAMPT [6] and which expresses CCR5 (Figure 6A). In this system, Rantes was, as expected, able to induce a Ca^2+^-response, albeit very small compared to HeLa-CCR5, that was blunted by the incubation with eNAMPT (Figure 6B,C).

Given that Rantes has been shown to promote migration, a Ca^2+^-dependent phenomenon, in melanoma cells [25], we next decided to investigate this via the wound-healing assay. Rantes, at a concentration of 100 ng/mL (12 nM) promoted wound closure after 24 h compared to control, in analogy to CCL3 (100 ng/mL = 9.9 nM). eNAMPT (500 ng/mL = 9 nM) per se had a small not significant effect on the migration of these cells. When eNAMPT was combined with Rantes or CCL3, it antagonized their effects on migration in the same way as maraviroc or the antagonist CCL7 (Figure 6D).

## 4. Discussion

The aim of the present work was to characterize the biological significance of the interaction between eNAMPT and CCR5 in cancer cells. In the present contribution, we show that (i) eNAMPT competes with Rantes for fluorescent binding in CCR5 overexpressing cells; (ii) eNAMPT, unlike Rantes, is unable to activate either the ERK or Ca^2+^-signaling pathways; (iii) pretreatment with eNAMPT results in the inhibition of CCR5 internalization and PKC activation, but not of ERK phosphorylation; (iv) eNAMPT antagonizes Rantes-dependent calcium signaling; (v) the effects of eNAMPT can be reproduced by eNAMPT H247E, the enzyme-dead mutant protein; and (vi) the effect on Ca^2+^-signaling is observable also in a melanoma cell line (B16) expressing endogenous levels of CCR5 and this most likely is linked to an inhibition of Rantes-induced migration by eNAMPT. Overall, therefore, we take our results to demonstrate that eNAMPT is an antagonist of some of the signaling cascades of CCR5.

On the other hand, eNAMPT has been shown in cell cultures to elicit effects that cannot be reconducted to inhibition of CCR5. For example, it activates STAT3, NF-κB, Akt, and p38 in the absence of CCR5 ligands [6,11]. It is therefore unlikely that CCR5 is the only plasma membrane receptor for eNAMPT. Indeed, two separate papers have now proven that eNAMPT is an agonist of the TLR4 receptor [10,11]. Furthermore, in the present manuscript, we solely looked at the interaction between eNAMPT and CCR5, as a binding K_D_ had already been ascertained [8], but we cannot exclude an involvement of other chemokine receptors in eNAMPT pathways. 

To add to the complexity, it has been recently demonstrated that the effects on pancreatic beta cell functional mass of eNAMPT are bimodal and concentration and structure functionally dependent [23]. In our study, we used concentrations between 250 and 1000 ng/mL (corresponding to 4.5–18 nM), with the only exception of internalization and cell labelling experiments, in which higher concentrations were used to counteract the higher concentrations used of RANTES (in the second instance, RANTES induces rapid internalization only at high concentrations, while in the first instance RANTES-PE binding could be appreciated only at higher concentrations). It is possible, therefore that at lower concentrations or higher concentrations than those tested in the present study eNAMPT interacts and engages with other receptors. In the aforementioned study, part of the concentration-dependence was attributed to a different effect of the monomer (enzymatically inactive), compared to the dimer. Given that monomeric mutants exist, it is likely that wider concentration-response curves should be performed to disclose bimodal actions and that both wild-type and mutant monomers should be employed. 

It is interesting that a natural antagonist of CCR5, CCL7 [26], like eNAMPT, also abolishes CCR5-dependent calcium signaling and cell migration. Surprisingly*, in silico* analysis using PyMol and Uniprot software reveals a common structure conformation and conserved sequence between eNAMPT and CCL7. As shown in Appendix A, the Pymol analysis of the NAMPT (cyano) and CCL7 (orange) structures showed a high homology. Specifically, CCL7 random coil (highlighted by the red square), which is used by the protein to bind to CCR5, is superimposable with a portion of NAMPT structure (amino acids 420–430). Moreover, the chemical-physical features of the amino acids of this portion are comparable to the steric hindrance of CCL7 in the same region (Appendix A). Given that CCL7 has been shown to be a promiscuous ligand, which alongside being a CCR5 antagonist is also an agonist of CCR1, CCR2, and CCR3, these receptors should be evaluated in the future to understand whether the effects of eNAMPT are the result of multiple interactions with a diverse array of receptors.

## Figures and Tables

**Figure 1 cells-09-00496-f001:**
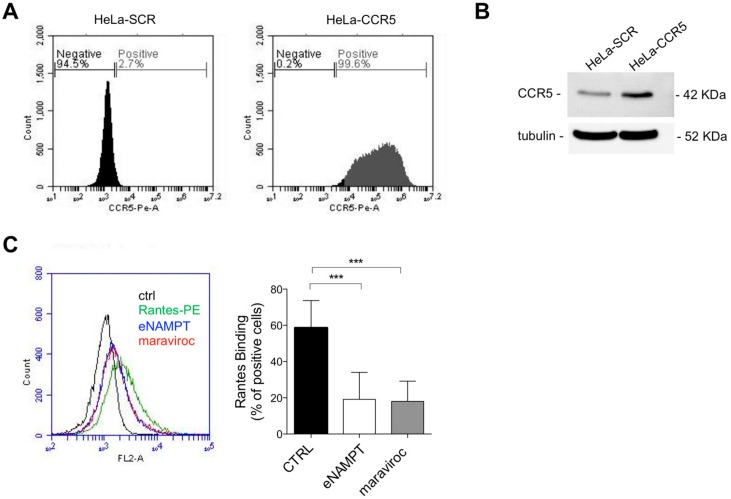
Extracellular nicotinamide phosphoribosyltrasferase (eNAMPT) binds to C-C chemokine receptor type 5 (CCR5) in HeLa cancer cells. (**A**) Representative flow cytometry analysis of CCR5 expression in HeLa-SCR and HeLa-CCR5 cells, using Rat anti-mouse CCR5 antibody. (**B**) Western blot analysis of CCR5 expression in HeLa-SCR and HeLa-CCR5 cells. The CCR5 antibody recognizes both human endogenous CCR5 and murine exogenous CCR5. (**C**) Representative FACS analysis and calculated percentage of positive cells of Rantes-PE (16 nM) binding on HeLa-CCR5 cells incubated in the presence or absence of eNAMPT (2.25 µM) or maraviroc (10 µM). Mean ± S.E.M. of five separate experiments; *** *p* < 0.001.

**Figure 2 cells-09-00496-f002:**
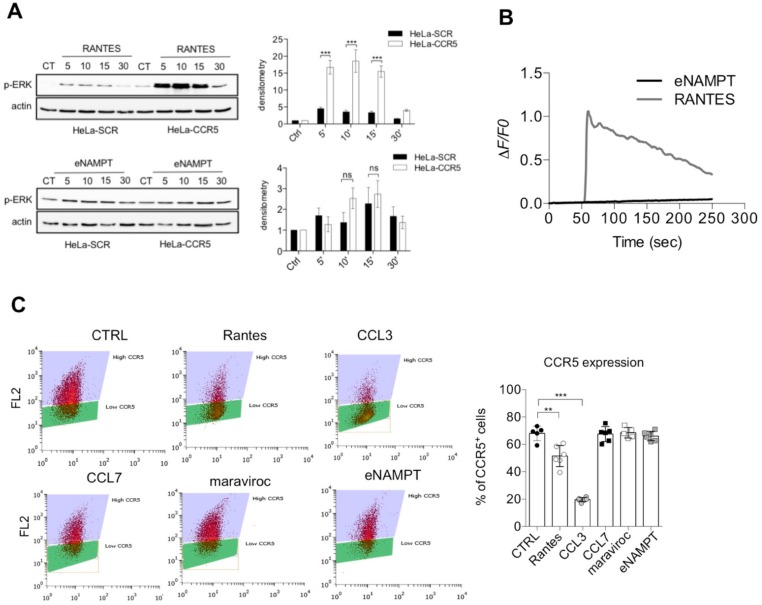
eNAMPT is not an agonist of CCR5. (**A**) Representative Western blot and densitometry analysis of phosphorylated ERK after 2 h of starvation followed by treatment for 5–30 min with recombinant Rantes (25 ng/mL; 3 nM) or eNAMPT (500 ng/mL; 9 nM) in serum-free conditions. Data from four separate experiments. (**B**) Representative calcium traces of HeLa-CCR5 loaded with FURA-2AM and stimulated with Rantes (25 ng/mL) or eNAMPT (500 ng/mL). Representative traces of 98–110 cells from 5–7 independent experiments. (**C**) Flow cytometry analysis of surface expression of CCR5 in HeLa-CCR5 cells treated for 1 h with Rantes (100 ng/mL = 12 nM), CCL3 (100 ng/mL = 9.9 nM), CCL7 (250 ng/mL = 22 nM), maraviroc (10 µM), and eNAMPT (2.5 µg/mL= 45 nM). The graph shows the mean ± S.E.M. of 12 determinations from four separate experiments. ** *p* < 0.01 *** *p* < 0.001; ns not statistically significant.

**Figure 3 cells-09-00496-f003:**
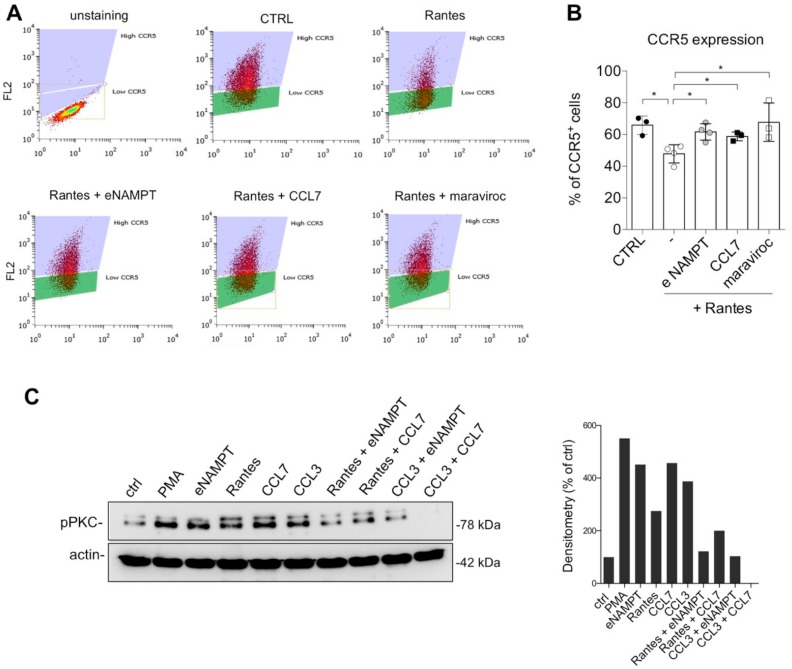
eNAMPT reduces CCR5 internalization and prevents pPKC activation. (**A**) Representative flow cytometry analysis of CCR5 expression in HeLa-CCR5 cells after the indicated stimuli. (**B**) Flow cytometry analysis of surface expression of CCR5 in HeLa-CCR5 cells treated for 1 h with Rantes (100 ng/mL; 12 nM) alone or pretreated for 20 min with eNAMPT (2.5 µg/mL; 45 nM) or CCL7 (250 ng/mL; 22 nM) or maraviroc (10 µM). Mean ± S.E.M. of four separate experiments. * *p* < 0.05. (**C**) Representative Western blot and densitometry of two independent experiments of phosphoPKC (pPKC) in Hela-CCR5 cells treated in serum-free for 20 min with Rantes (25 ng/mL; 3 nM), eNAMPT (500 ng/mL; 9 nM), CCL7 (250 ng/mL; 22 nM), or CCL3 (100 ng/mL; 9.9 nM) alone or combined.

**Figure 4 cells-09-00496-f004:**
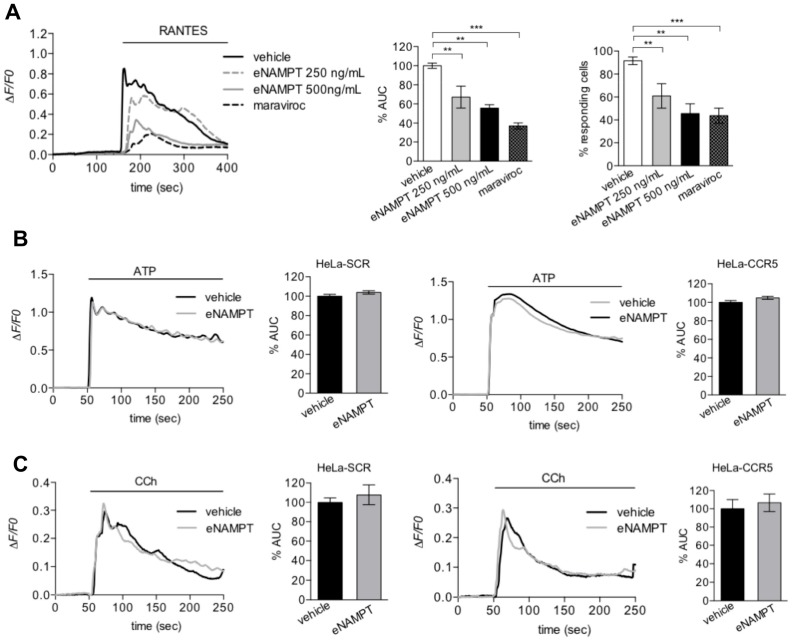
eNAMPT inhibits Rantes-induced cytosolic Ca2+-signals. (**A**) Representative calcium traces of HeLa-CCR5 loaded with FURA-2AM and treated with vehicle, eNAMPT (250 ng/mL; 4.5 nM or 500 ng/mL = 9 nM), or maraviroc (10 µM) 100 s before the addition of 25 ng/mL (3 nM) of Rantes. Histograms of responding cells (middle panel) and percentage of Area Under the Curve ( AUC )(right panel) as mean ± S.E.M. (248–410 cells from 6–11 independent experiments). (**B**,**C**) Representative calcium traces and percentage of AUC of HeLa-SCR and HeLa-CCR5 loaded with FURA-2AM and treated with 3 µM of ATP (**B**) or 300 µM of CCh (**C**) alone or pretreated with eNAMPT (500 ng/mL = 9 nM) for 5 min. The data are summarized in histograms and expressed as mean ± S.E.M of 120–190 cells (from 5–9 independent experiments) and 105–185 cells (from 5–9 independent experiments), respectively. ** *p* < 0.01 *** *p* < 0.001.

**Figure 5 cells-09-00496-f005:**
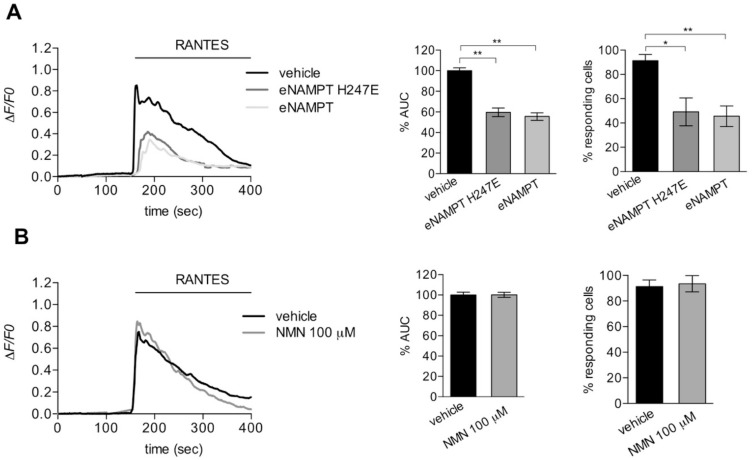
The catalytic activity of eNAMPT is dispensable for CCR5 antagonism. (**A**) Left, representative calcium traces of HeLa-CCR5 loaded with FURA-2AM and treated with RANTES (25 ng/mL; 3 nM) alone or pretreated with eNAMPT/eNAMPT H247E (500 ng/mL; 9 nM) for 100 s. Right, histograms of percentage of AUC (middle panel) and responding cells (right panel) as mean ± S.E.M. (180–210 cells from four independent experiments). (**B**) Left, representative calcium traces of HeLa-CCR5 loaded with FURA-2AM and pretreated for 100 s with vehicle or NMN (100 µM) before the addition of Rantes (25 ng/mL = 3 nM). Histograms of percentage of AUC (right panel) and responding cells (left panel) as mean ± S.E.M. (99–126 cells from three independent experiments). * *p* < 0.05, ** *p* < 0.01.

**Figure 6 cells-09-00496-f006:**
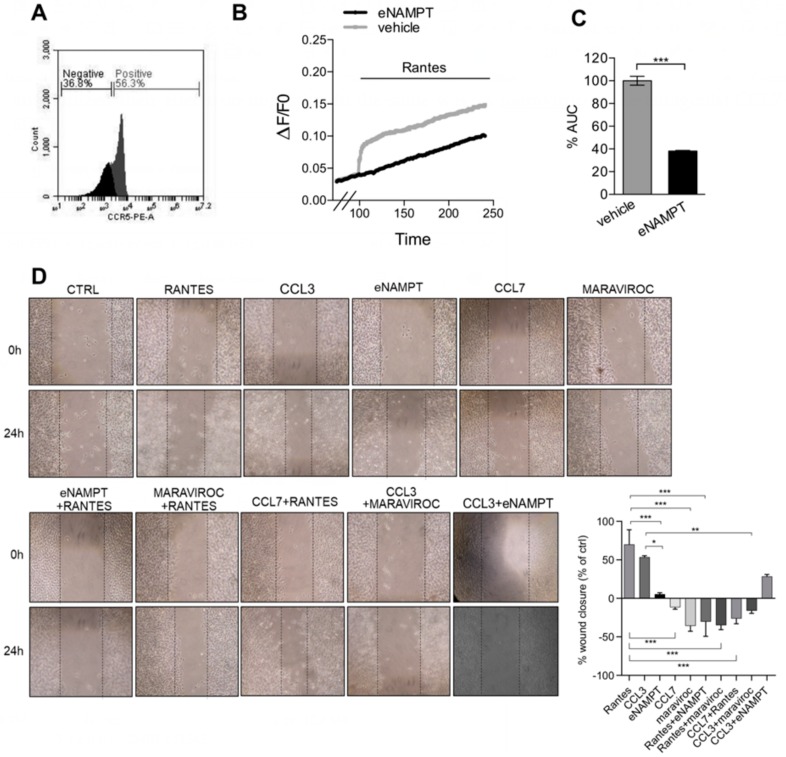
eNAMPT antagonizes the CCR5-mediated migration of B16 melanoma cells. (**A**) CCR5 expression in B16 cells stained with anti-mouse CCR5-PE and analyzed by flow cytometry. (**B**,**C**) AUC and percentage of responding cells of B16 cells loaded with FURA-2AM and pretreated with eNAMPT (500 ng/mL; 9 nM) for 100 s before the addition of RANTES (100 ng/mL; 12 nM) at 100 s. Mean ± S.E.M. of 180–210 cells from four independent experiments. (**D**) Left, representative wound healing images of B16 cells treated with vehicle, Rantes (100 ng/mL; 12 nM), eNAMPT (500 ng/mL; 9 nM) and/or maraviroc (10 µM), and/or CCL7 (250 ng/mL; 22 nM) and CCL3 (100 ng/mL; 9.9 nM) at time 0 and after 24 h of treatment. Right, percentage of wound closure (compared to % of the control) after 24 h of treatment. Mean ± S.E.M. of six determinations from two separate experiments. * *p* < 0.05, ** *p* < 0.01, *** *p* < 0.001.

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
