# Peer review of "The Cytokine Nicotinamide Phosphoribosyltransferase (eNAMPT; PBEF; Visfatin) Acts as a Natural Antagonist of C-C Chemokine Receptor Type 5 (CCR5)"

_cells, 2020, doi:10.3390/cells9020496_

Round 1

Reviewer 1 Report

This is an interesting study that is highly relevant and important to the field.

The existence and identity of eNAMPT receptors is of major importance to the field and this study significantly advances that through a range of experimental approaches, ultimately identifying that eNAMPT at high concentrations antagonises the effects of CCR5.

My main issue with the study is the concentration of eNAMPT which was chosen for these studies. Firstly, I am unclear why the concentrations used vary so much. e.g. approx 500 ng/ml for most experiments, but 2.5 ug/ml for internalisations experiments? Please also provide a conversion for ng/ml to uM for eNAMPT - i.e. it is not clear how the stated eNAMPT affinity of 5uM for CCR5 relates to the ng/ml concentrations used in these experiments.

Second - what is the rational for using concentrations of up 250-1000 ng/ml, other than these concs having been used in some previous studies? Serum levels of eNAMPT are often measured much lower than this - either in the low ng/mL or roughly 30 - 100 ng/ml levels depending on which assay kit is used (Adipogen kit seems to give lower values, possibly due to off target effects of the Pheonix peptide kit). This is important since very recent studies (Sayers et al, Diabetologia, 2019) have reported bi-modal effects of eNAMPT, with lower concentrations exerting distinct effects (including on calcium levels) when compared to higher concentrations.

Have the authors analysed the effects of lower ng/mL concentrations of eNAMPT on CCR5? Given the reports by Sayers et al, it is possible that an entirely different effect on CCR5 may be observed at lower concentrations.

The recent study by Yoshida et al (Cell Metab) which showed eNAMPT circulating in extra-cellular vesicles should be mentioned in the introduction.

Reviewer 2 Report

see enclosed file attached 

Reviewer 3 Report

The cytokine Nicotinamide 2 Phosphoribosyltransferase (eNAMPT; PBEF; visfatin) acts as a natural antagonist of C-C chemokine receptor type 5 (CCR5) by Torretta, Colombo et al.

In this paper the authors discuss the interesting functional interaction between eNAMPT and CCR5. They showed that eNAMPT can act as natural antagonist of CCR5 for several reasons summarized in the discussion section: (i) eNAMPT competes with Rantes for fluorescent binding in CCR5 over-expressing cells; (ii) eNAMPT, unlike Rantes, is unable to activate either the ERK or Ca2+-signaling pathways; (iii) the pre-treatment with eNAMPT results in the inhibition of CCR5 internalization and PKC activation, but not of ERK phosphorylation; (iv) eNAMPT antagonizes Rantes-dependent calcium signaling. Moreover, they showed that this antagonistic activity is independent from its enzymatic activity, as reported using eNAMPT H247E, the enzyme-dead mutant protein.

Major points of the work:

Figure 1: I would like to see a western blot of CCR5-overexpressing cells Figure 2A: ERK activation is reported in several papers after NAMPT treatment. In the blot (actin is not perfectly comparable in all lanes) the increase of p-ERK is visible upon NAMPT exposure, without different in HELA-SCR and HELA-CCR5. This suggests that the activation of this pathway is independent from eNAMPT/CCR5 binding. In the legend the time of treatment is 5 minutes, but in the blot there are 5-10-15-30 (are different time points?). Please correct the legend. Related to this, Figure S2 showed that eNAMPT was unable to interfere with Rantes-mediated ERK activation, but the point here is: Why is there pERK activation by eNAMPT in HELA-SCR but not in HELA-CCR5? This is different compared the blot shown in Figure 2A. MAPK activation by eNAMPT seems to be a pathway not related to the expression of CCR5. Are HELA cells positive for TLR4 expression? I think that the authors should be discuss these data, and uniform the results of the blots. In Figure 4 and in the corresponding result, the authors “stressed” that the effects of eNAMPT are specific, but the profiles of buffers, NMNAT and ADH were not shown. I suggest to put them in a supplementary Figure. Why did the authors perform only Ca2+ signaling experiments with mutant H247E? I would like to see also CCR5 internalization and PKC activation. Regarding the melanoma cell line model the authors in the discussion wrote that (vi) the effect on Ca2+-signaling is observable also in a melanoma cell line expressing endogenous levels of CCR5 (B16) and this most likely is linked to an inhibition of Rantes-induced migration by eNAMPT. In my opinion the wound-healing experiments are not well presented, if the authors would put this part in the paper additional experiments should be added (not only 2). The images in the Figure are at very low resolution, and moreover I think that eNAMPT alone promotes wound repair (there is an evident different between T0 and T24), maybe not via CCR5. The size of the wounds in T0 images is different. How did the authors calculate the repair index? The inhibition of Rantes-induced migration by eNAMPT is debatable. Is there any statistical significant in the cumulative histogram? I suggest to see also invasion in matrigel, a second readout to investigate the migration potential of the cells. Why did not the authors repeat the same experiments (CCR5 internalization, PKC activation) also in B16 model? Migration of a cancer cell is a complex process, maybe is more difficult to see statistical significant differences. This part on B16 is a weak point of the paper.

Round 2

Reviewer 1 Report

All of my queries on the original manuscript have been addressed

Reviewer 2 Report

Dear Co-authors:

I am very happy with the revisions you have made to this manuscript entitled "the cytokine eNAMPT acts as a natural antagonist of C-C chemokine receptor type 5 (CCR5)".  Thus, I recommend that it could be publish in the current format.

Best,

Liliana Moreno PhD

Reviewer 3 Report

I'm satisfied about the new experiments added and the revision of the text.